

# Charcot-Marie-Tooth type 4B2 demyelinating neuropathy in miniature Schnauzer dogs caused by a novel splicing *SBF2 (MTMR13)* genetic variant: a new spontaneous clinical model

Nicolas Granger[1,2], Alejandro Luján Feliu-Pascual[3], Charlotte Spicer[4], Sally Ricketts[5], Rebekkah Hitti[5], Oliver Forman[5], Joshua Hersheson[4] and Henry Houlden[4]

[1] Royal Veterinary College, University of London, Hatfield, United Kingdom
[2] Bristol Veterinary Specialists, CVS Referrals, Bristol, United Kingdom
[3] Neurology/Neurosurgery Service, Aúna Especialidades Veterinarias, Valencia, Spain
[4] Department of Molecular Neuroscience, UCL Institute of Neurology & National Hospital for Neurology and Neurosurgery & London, London, United Kingdom
[5] Kennel Club Genetics Centre, Animal Health Trust, Newmarket, United Kingdom

Corresponding author
Henry Houlden, h.houlden@ucl.ac.uk

## ABSTRACT

**Background.** Charcot-Marie-Tooth (CMT) disease is the most common neuromuscular disorder in humans affecting 40 out of 100,000 individuals. In 2008, we described the clinical, electrophysiological and pathological findings of a demyelinating motor and sensory neuropathy in Miniature Schnauzer dogs, with a suspected autosomal recessive mode of inheritance based on pedigree analysis. The discovery of additional cases has followed this work and led to a genome-wide association mapping approach to search for the underlying genetic cause of the disease.

**Methods.** For genome wide association screening, genomic DNA samples from affected and unaffected dogs were genotyped using the Illumina CanineHD SNP genotyping array. *SBF2* and its variant were sequenced using primers and PCRs. RNA was extracted from muscle of an unaffected and an affected dog and RT-PCR performed. Immunohistochemistry for myelin basic protein was performed on peripheral nerve section specimens.

**Results.** The genome-wide association study gave an indicative signal on canine chromosome 21. Although the signal was not of genome-wide significance due to the small number of cases, the *SBF2* (also known as *MTMR13*) gene within the region of shared case homozygosity was a strong positional candidate, as 22 genetic variants in the gene have been associated with demyelinating forms of Charcot-Marie-Tooth disease in humans. Sequencing of *SBF2* in cases revealed a splice donor site genetic variant, resulting in cryptic splicing and predicted early termination of the protein based on RNA sequencing results.

**Conclusions.** This study reports the first genetic variant in Miniature Schnauzer dogs responsible for the occurrence of a demyelinating peripheral neuropathy with abnormally folded myelin. This discovery establishes a genotype/phenotype correlation in affected Miniature Schnauzers that can be used for the diagnosis of these dogs. It further supports the dog as a natural model of a human disease; in this instance,

Charcot-Marie-Tooth disease. It opens avenues to search the biological mechanisms responsible for the disease and to test new therapies in a non-rodent large animal model. In particular, recent gene editing methods that led to the restoration of dystrophin expression in a canine model of muscular dystrophy could be applied to other canine models such as this before translation to humans.

## INTRODUCTION

Charcot-Marie-Tooth (CMT) disease is the most common neuromuscular disorder of the human population, affecting 40 out of 100,000 individuals in Europe and the United States (*Skre, 1974*; *Pareyson & Marchesi, 2009*; *Reilly, Murphy & Laura, 2011*) and more than 2.6 million people worldwide (*Emery, 1991*). It was jointly described by Drs. Charcot and Marie in Paris and Dr. Tooth in Cambridge in 1886 (*Charcot, 1886*). It is now well established that CMT disease refers to a wide group of degenerative motor and sensory neuropathies, 70% of which are hereditary (*Pareyson & Marchesi, 2009*; *Vallat, Mathis & Funalot, 2013*; *Houlden & Reilly, 2006*). More than 60 genes encompassing ∼900 genetic variants have been implicated in the pathogenesis of CMT (*Bouhy & Timmerman, 2013*) offering the potential for further research to elucidate alterations in biological cascades causing the neuropathies. However, few advances have been made so far and CMT remains a slowly progressive and incurable disease. Current research is focused on the use of rodent transgenic models, which have several limitations. They are difficult and time consuming to obtain and the expected phenotype cannot be guaranteed (*Bouhy & Timmerman, 2013*), limitations that are not present with spontaneously occurring animal models.

In companion dogs, CMT-like diseases naturally occur, as evidenced by the phenotypic descriptions made by veterinarians in 22 breeds up until 2011 (*Granger, 2011*; *Coates & O'Brien, 2004*). We have now entered a phase where veterinary researchers and canine geneticists are taking advantage of next-generation sequencing techniques in order to identify the molecular defects underlying these diseases in dogs, sometimes with a very small number of affected individuals (*Forman, De Risio & Mellersh, 2013*; *Fenn et al., 2016*). This promises to lead to an exponential unravelling of genetic variants in animals spontaneously affected by inherited neurological diseases.

In support of this, in the last decade, several variants within five different genes have been reported in eight canine breeds affected by inherited polyneuropathies similar to CMT disease in humans. First found in 2009, a deletion in the mitochondrial *tRNATyr* gene in Golden Retrievers causes a sensory neuropathy (*Baranowska et al., 2009*). Following on, variants of the *NDRG1* gene, implicated in myelination, cause an early-onset progressive polyneuropathy in Alaskan Malamutes (*Bruun et al., 2013*) and a hereditary motor and sensory polyneuropathy in Greyhounds (*Drogemuller et al., 2010*).

The *ARHGEF10* gene, involved in neuronal growth and axonal migration, is mutated in some Leonberger and Saint Bernard dogs with severe juvenile-onset polyneuropathy and laryngeal paralysis (*Ekenstedt et al., 2014*) whereas a variant of the *GJA9* gene, which is part of the connexin gap junction family proteins, causes a late onset polyneuropathy with laryngeal paralysis in Leonberger dogs (*Becker et al., 2017*). Finally, the *RAB3GAP1* gene, involved in Rab proteins' function in membrane trafficking in the endoplasmic reticulum, axonal transport, autophagy and synaptic transmission, has a single nucleotide deletion in Black Russian terrier dogs (*Mhlanga-Mutangadura et al., 2016b*) and a SINE insertion in Alaskan Huskies (*Wiedmer et al., 2015*), leading to juvenile-onset, laryngeal paralysis, polyneuropathy and ocular changes similar to Warburg micro syndrome in humans. In Rottweilers, a variant in the same *RAB3GAP1* gene causes neuronal vacuolation and spinocerebellar degeneration (*Mhlanga-Mutangadura et al., 2016a*). Older neuropathies, such as the well-known inherited sensory and autonomic in the Border Collie first described in 1987 (*Wheeler, 1987*), have also benefited from genetic advances. In 2016 one causal genetic variant of *FAM134B*, which encodes a *cis*-Golgi protein found in sensory and autonomic ganglion neurons, has been discovered (*Forman et al., 2016*) and two-years later, this mutation was also described in two mixed breed dogs (*Amengual-Batle et al., 2018*). In four other breeds, the German short-haired Pointer, the English Pointer, the English Springer Spaniel and the French Spaniel, some known since 1964, a similar inherited sensory and autonomic exist and found to be linked to a genetic variation of the long-non-coding RNA located upstream of the *Glial cell-Derived Neurotrophic Factor* gene, coding for a neurotrophic factor involved in neuronal development and adult neuronal survival (*Plassais et al., 2016*). The discovery of this genetic variant is particularly relevant to hereditary sensory and autonomic neuropathies in humans where it is not known.

In 2008, we described the clinical, electrophysiological and pathological findings of a demyelinating hereditary motor and sensory neuropathy in Miniature Schnauzer dogs, characterized by the presence of focally folded myelin sheaths also known as *tomacula* and segmental demyelination (*Vanhaesebrouck et al., 2008*). After exclusion of variants in the *PMP22* and *P0* myelin genes by direct sequencing, the aim of this study was to demonstrate that the demyelinating hereditary motor and sensory neuropathy in Miniature Schnauzer dogs was genetic in origin by conducting a genome-wide association study (GWAS) and resequencing of candidate genes. From these investigations, a splice variant in the *SBF2* (SET-binding factor 2), also known as *MTMR13* (myotubularin-related protein-13) gene was identified. We will be using the official HUGO Gene Nomenclature Committee gene name *SBF2* throughout the remainder of the manuscript to refer to the *SBF2/MTMR13* gene.

## MATERIALS & METHODS

### Study samples

All dogs were examined and investigated by veterinary neurologists. Blood was collected from Miniature Schnauzers pre-mortem and tissue samples were collected post-mortem after dogs had been euthanized on welfare grounds. Euthanasia was carried out solely to
alleviate suffering and no healthy individuals were sacrificed for use in this study. Euthanasia was carried out in accordance with the Veterinary Surgeons Act 1966 and under the auspices of the RCVS. All samples used in this study were collected after permission had been granted by dog owners (24-2018E).

## Sample DNA and RNA extraction

All DNA samples were collected from privately owned pet dogs by blood or tissue extraction using the QIAamp midi or mini DNA extraction kit (Qiagen). Affected dogs are described below and some have been previously reported. Controls were Miniature Schnauzer dogs that were reported as clinically healthy with no sign of neuropathy based on our neurological examination of these cases ($n = 7$), or apparently healthy Miniature Schnauzers with no owner report of neurological disorder ($n = 224$). Cerebellum or muscle tissue samples were collected post-mortem and DNA and total RNA extracted using the RNeasy midi kit (Qiagen). Genomic DNA from AHT control dogs utilized for this study were derived from buccal swab samples or from residual blood drawn for diagnostic veterinary purposes as part of the dog's veterinary care. Samples were taken following informed and written owner consent. Sample collection for genetic research has been approved by the Animal Health Trust Ethics Committee (24-2018E).

Total RNA was extracted from homogenized muscle from an affected dog with two copies of the *SBF2* variant and a control dog using a miRNeasy Mini Kit (Qiagen, Chatsworth, CA) according to the manufacturer's instructions. First-strand cDNA synthesis was then carried out using SuperScript II Reverse Transcriptase First-Strand Synthesis System for RT-PCR (Invitrogen/Life Technologies). The synthesizing conditions included initial 5 minutes' incubation at 65 °C with dNTP and a mix of random hexamers/oligo dT primers, after which tubes were placed on ice for at least 1 min. cDNA synthesis mix ($5 \times$ First-Strand Buffer, 0.1M DTT, RNase OUT) was added and incubated at 25 °C for 2 min. SuperScript II was added and tubes incubated at 25 °C for 10 min, followed by 42 °C for 50 min. The reaction was terminated by incubation at 70 °C for 5 min.

## GWAS analysis

Genomic DNA samples from the first two Miniature Schnauzers diagnosed with a peripheral neuropathy in France by one author (NG), and 39 Miniature Schnauzer controls were genotyped using the Illumina CanineHD SNP genotyping array that comprises 173,662 SNPs (i.e., a total of 41 dogs). The SNP genotyping dataset was analyzed for association using the statistical package PLINK (*Purcell et al., 2007*). As part of the 41 dogs, a set of nine dogs (comprising the two affected cases and seven controls) and a set of 32 controls were genotyped by two centers; University College London (JH, HH) and Animal Health Trust (SLR, RJH, OPF) respectively. To preserve data quality SNP QC was conducted in each dataset independently before merging. Sample call rates for all individuals were >99%. SNP quality control filtering excluded SNPs with a minor allele frequency of less than 5% and a genotyping call rate of less than 97% in each independent dataset. This resulted in 74,374 SNPs for association analysis in the merged dataset. Multidimensional scaling analysis did not indicate the presence of population stratification between the two sets (File S1). Later,

a further 192 Miniature Schnauzers (Animal Health Trust) with no report of the disorder were included as a follow-up set along with six other affected cases (five from Spain, one from Belgium), altogether leading to a set of eight affected dogs and 231 controls.

Due to the very small sample size and disproportionate case-control ratio we conducted the GWAS analysis using a Fisher's exact test with 100,000 permutations (max(T) permutation procedure) to account for multiple testing. (The threshold for genome-wide statistical association after permutation is 1.3.)

### *SBF2* resequencing

This was done using the two originally described French cases; five affected cases available from Spain; and one case from Belgium –these dogs presented with similar neurological and electrophysiological signs as originally described. Genomic DNA was amplified using a standard touchdown PCR reaction. Cycling conditions were as follows: 94 °C (5 min); 94 °C (30 s), 60 °C (30 s), 72 °C (45 s) for 25 cycles; 94 °C (30 s) 50 °C (30 s) 72 °C (45 s) for 13/18 cycles, 72 °C (10 min). Primers for each of 35 coding exons with flanking intronic regions were designed with PrimerZ (http://ncbi36.genepipe.ncgm.sinica.edu.tw/primerz/beginDesign.do) (see sequences in File S2). Enzymatic clean-up with Exonuclease I and *FastAP* Thermosensitive Alkaline Phosphatase (Thermo Scientific) was carried out and subsequent sequencing performed. PCR primers were used for sequencing along with BigDye Terminator v.1.1 cycle sequencing kit (Applied Biosystems). Cycling conditions were as follows: 94 °C (1 min); 94 °C (30 s), 50 °C (15 s), 60 °C (4 min) for 25 cycles. Sequencing reaction products were purified using sephadex columns and then run on a 3730xl DNA Analyzer. The resulting data sequences were aligned to the canine *SBF2* reference sequence (CanFam 3.1) retrieved from Ensembl (ENSCAFG00000007500; ensembl.org) and manually analyzed using Sequencher v. 4.9 (Gene Codes Corporation, Ann Arbor, MI, USA). Gene Runner was used to investigate the amino acid sequence.

### Variant genotyping and statistical analysis

The *SBF2* variant was genotyped using an allelic discrimination assay using an ABI StepOne real-time thermal cycler. Primers and allelic discrimination probes were designed using Primer3 (*Untergasser et al., 2012*) and obtained from Integrated DNA Technologies (IDT). Primer sequences were: TGGGAGAGTGGAAGCAACAG (forward); GGAGTGTCTCTG-TATGTGCACATT (reverse); 5′-FAM- TTGTCACAAACAGTTACCAA-NFQ-3′ (mutant); 5′-VIC- TTGTCACAAACAGGTACCAA-NFQ-3′ (wildtype). Individual probe assays were re-suspended in ultrapure water to a 40X mix and combined. Allelic discrimination reactions were prepared using KAPA probe fast (KAPA Biosystems) with the 40X probe mix. Thermal cycling conditions were: 25 °C for 30 s; 95 °C for 3 min; followed by 40 cycles of 95 °C for 3 s; and 60 °C for 10 s, then the reaction held at 25 °C for 30 s.

As the variant was rare in our sample set, a subset of genotypes was verified by Sanger sequencing. Primer sequences were AACCTCATGGTCAACCTGCT (forward) and TGCACAGCAGTATTTGCCTAA (reverse). PCR cycling consisted of an initial 95 °C denaturation step for 5 min, followed by 35 cycles of 95 °C for 30 s; 60 °C for

30 s; and 72 °C for 1 min. Final elongation was 72 °C for 5 min. PCR products were purified using 96-well multiscreen PCR plates (Merk Millipore) and sequenced in both directions using BigDye Terminator v3.1 chemistry (Applied Biosystems) and an ABI 3130XL Genetic Analyzer (Applied Biosystems). Sequences were aligned and manually called using the Staden package (*Staden, 1996*).

The association between demyelinating neuropathy and genotype was assessed using the Fisher's exact test.

## Immunohistochemistry of peripheral nerve sections

Peripheral nerve specimens originating from surgical biopsies at the time of diagnosis were embedded in OCT and snap frozen in isopentane pre-cooled with liquid nitrogen and stored at −80 °C. Cryostat sections of 30 μm were then cut and mounted on histopathology slides and air dried for 30 min at 37 °C. The sections were stained for myelin basic protein as follows: incubation in a blocking solution of 10% normal goat serum 0.1M PBS and 0.3% t-octylphenoxypolyethoxyethanol for an hour at room temperature; incubation with primary antibody (anti-myelin basic protein monoclonal antibody, 1:100, Chemicon) diluted in 0.1M PBS containing normal goat serum overnight at 4 °C; washing 3 times with 0.1 M PBS for 5 min; incubation with a secondary antibody (Alex Fluor 488, goat anti-rat, ThermoFischer Scientific) for 2 h, in the dark, at room temperature. The sections were then examined under a fluorescent microscope.

## RESULTS

### Disease phenotype

The phenotype of this inherited demyelinating peripheral nerve disease was previously reported and affected dogs share familial links, based on the pedigree analysis of 32 dogs (see Figure. 2 in *Vanhaesebrouck et al. (2008)*). Briefly, at a young age (<2 years), affected Miniature Schnauzers presented regurgitations caused by mega-esophagus and inspiratory dyspnea caused by laryngeal paralysis. Electrophysiological studies revealed marked slowing of motor and sensory nerve conduction velocities (∼20 m/s), although some nerves had preserved conduction velocities. Although the prognosis remains guarded, long survival is observed with dogs alive >3 years following diagnosis. Teased nerve fibers presented variable thickness of the myelin sheath with characteristic '*tomacula*' and area of segmental demyelination. Toluidine-blue-stained semi-thick resin-embedded sections demonstrated loss of myelinated fibers, thinly myelinated fibers and redundant myelin loops again characteristic of the presence of *tomacula*, as previously reported by *Vanhaesebrouck et al. (2008)*. Muscle biopsy specimens appeared normal when sampled (*Vanhaesebrouck et al., 2008*). Immunohistochemistry of peripheral nerve sections stained for myelin basic protein (Fig. 1) demonstrate the loss of normal myelin architecture in affected cases.

### Mapping of the causative variant

The genome-wide association study was conducted using two cases originally described in 2008 (*Vanhaesebrouck et al., 2008*) and 38 apparently healthy breed-matched controls (one control sample did not work during testing). The strongest associated SNP was located on

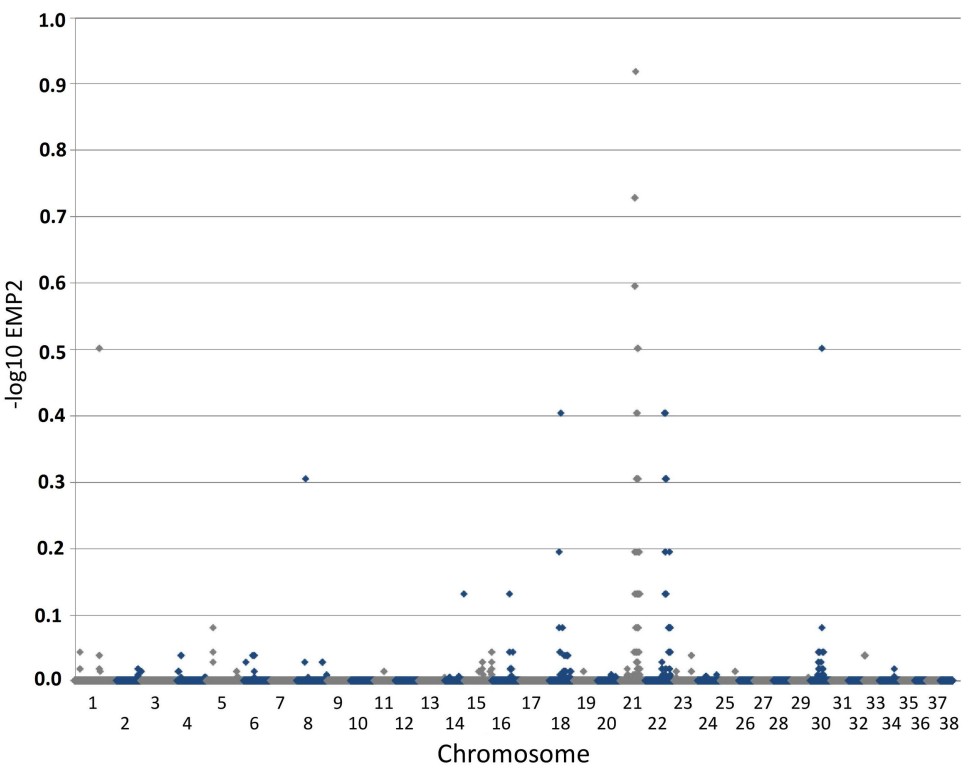

**Figure 2** **Manhattan plot toillustrate results of GWAS analysis.** Manhattan plot on two cases and 39 controls; 74,374 SNPs. The X-chromosome is omitted. Analyses were conducted using Fisher's exact test and 100,000 permutations to correct for multiple testing using the max(T) permutation procedure in PLINK. EMP2 corrected empirical *P*-value.

canine chromosome 21 (BICF2S2377579 –CanFam 3.1 position chr21: 34,045,084) (Fig. 2), although this did not reach genome-wide statistical association due to the small dataset and concurrent lack of statistical power. Interrogation of the SNP genotype distributions across the associated region identified a run of shared homozygosity in cases spanning from chr21:18,552,117-46,804,230 (CanFam3.1). The most obvious candidate gene in this region was the *SBF2* gene, which causes CMT in human patients with a clear phenotype.

Resequencing of the *SBF2* gene was initially performed using one of the two affected case and one control. This revealed a homozygous +1 splice genetic variant in exon 19 (c.2363+1 G>T; chr21:33,080,022 C>A CanFam3.1) in the affected case in comparison to the control (Fig. 3). No further variants were found in the entire coding sequence. This putative disease-causing c.2363+1 G>T variant was assessed in the initial GWAS set (2 affected and 39 controls) *plus* six additional affected cases and a further 192 Miniature Schnauzers with no report of the disorder, demonstrating a highly significant statistical association (see Table 1) in comparison to the top GWAS SNP. This identified 10 heterozygotes and 182 dogs with a homozygous wildtype genotype in the added 192 controls and 38 homozygous wildtype genotypes in the GWAS set (one control from this set did not work); but no dogs homozygous for the variant in the control set. The 6 affected cases,

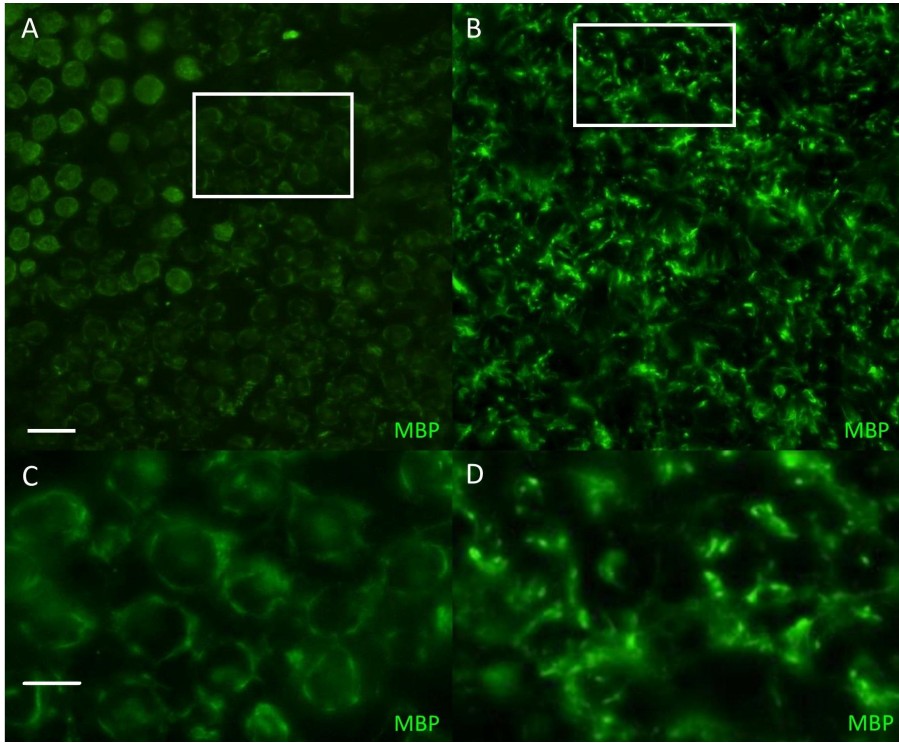

**Figure 1** **Immunofluorescent staining for myelin basicprotein in canine peripheral nerves.** (A, C) Transverse sections (30 μm) obtained from the tibial nerve of a normal Miniature Schnauzer dog stained for myelin basic protein (green)—note the preservation of the peripheral myelin architecture delineating axons within the fascicle; (B, D) Transverse sections (30 μm) obtained from the tibial nerve of an affected Miniature Schnauzer dog—the myelin has lost its organization. C and D are magnified images of A and B white rectangles respectively. Scale bar A, C = 20 μm; scale bar B, D = 0 μm.

5 from Spain and one from Belgium, were mutant homozygous dogs. Screening for the variant in a dataset of 802 whole genome sequences from 162 purebred dogs/mixed breed dogs/wolves identified two Miniature Schnauzers that were heterozygous for the variant and one mutant homozygote; the remaining 799 dogs were clear (File S3). On follow-up investigation the mutant homozygous dog was affected with neonatal lethal spondylocostal dysostosis, hence no clinical data relevant to the demyelinating neuropathy phenotype was available (*Willet et al., 2015*). The original two GWAS cases were found in France and the mutant homozygote identified through genome data was of Australian origin, although the sire was from Sweden. The additional six affected cases were from Spain and Belgium. This tentatively suggests presence of the variant in different regions of the globe.

## Effect of *SBF2* variant on mRNA splicing and coding sequence

To analyze the consequence of this splicing acceptor site variant on *SBF2* pre-mRNA splicing, RNA was extracted from dissected muscle of an unaffected and an affected dog (the case found in Belgium) and RT-PCR performed. The resulting transcripts were sequenced which revealed the presence of a 40-bp deletion at the 3′ site of exon 19 due to the induction of a cryptic splice site in this exon (Fig. 4A). The *SBF2* genetic variant and

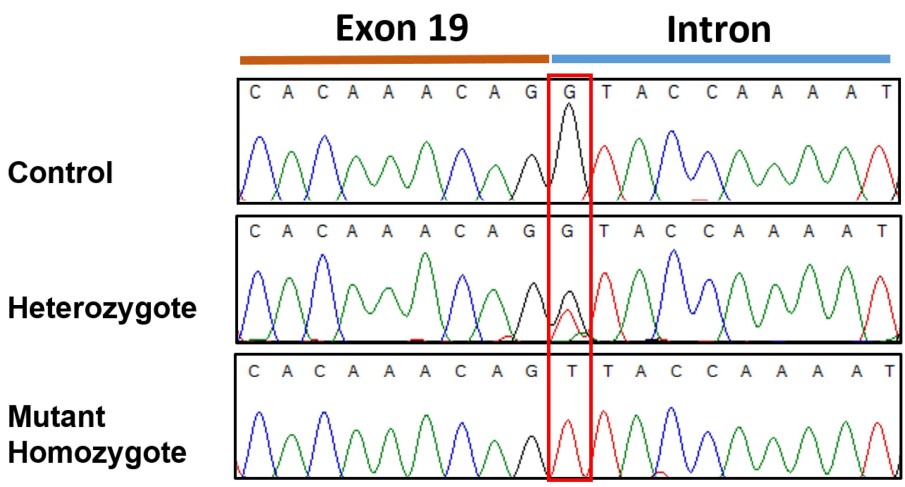

**Figure 3** **Sanger sequencingelectropherogram traces of the c.2363 +1 G > T variant (boxed) in** *SBF2* **exon 19.** Trace shows the 3′end of exon 19 and the adjacent intron from PCR products amplified from genomic DNA.

**Table 1** **Miniature Schnauzers genotyped for the candidate variant originating from the initial GWAS set (two affected cases and 38 controls[b]), six other affected cases and 192 controls.**

| c.2363+1 genotype[a] | Case (n) | Control (n) | Total | |
|---|---|---|---|---|
| G/G | 0 | 220 | 220 | |
| G/T | 0 | 10 | 10 | |
| T/T | 8 | 0 | 8 | *P*-value 4.4 $\times 10^{-15}$ |

**Notes.**
[a]T is mutant allele.
[b]NB Data on c.2363+1 genotype is missing for one control dog.

resulting 40-bp deletion leads to a premature stop codon (p.G775V$fs$ *5) and truncates the protein by 1070 amino acids (Fig. 4B).

## DISCUSSION

We previously described a suspected inherited demyelinating peripheral neuropathy in Miniature Schnauzer dogs and highlighted its similarities with some demyelinating forms of CMT in humans (*Vanhaesebrouck et al., 2008*; *Niemann, Berger & Suter, 2006*). Mainly, we discovered in the dog the presence of specific neuropathological features of the peripheral nerves, such as abnormal folding of peripheral myelin around the axons termed *tomacula*, as originally described in humans with certain demyelinating neuropathies (*Ohnishi et al., 1989*; *Gabreels-Festen et al., 1990*). An inherited trait for the canine disease equivalent was suggested based on pedigree analysis of affected dogs. Our findings led us to suspect that variants of genes coding for: (i) myelin proteins (such as *PMP22*, *P0*, *periaxin*); (ii) intracellular Schwann cell proteins involved in the synthesis, transport and degradation of myelin proteins including the myotubularin-related (MTMR) proteins; and (iii) regulating myelin gene transcription proteins (such as *EGR2*), might

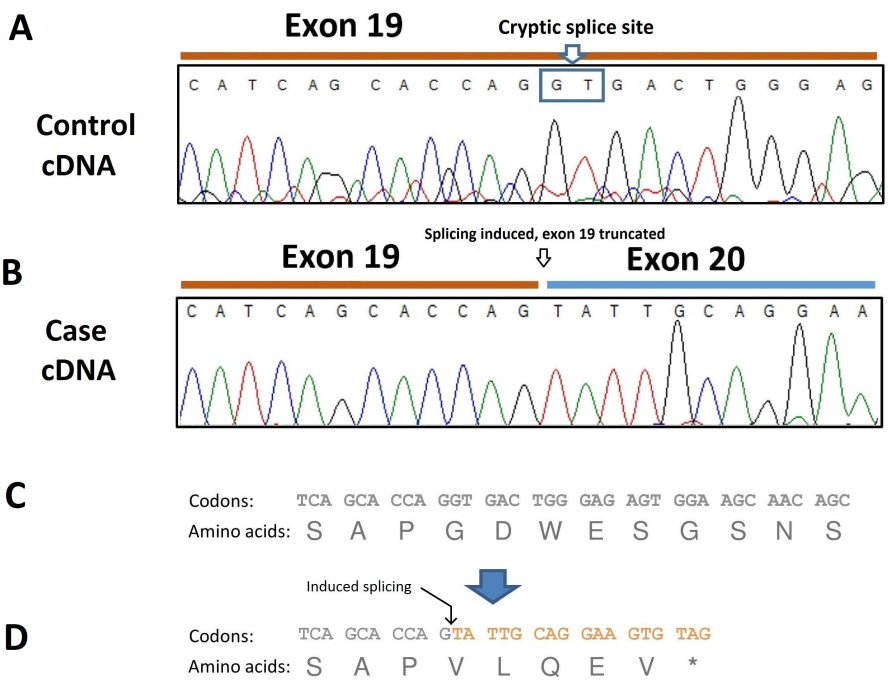

**Figure 4 Electropherogram of PCR-amplified cDNA of control and affected dogs.** (A) Electrophero-gram of PCR-amplified cDNA from a control dog; (B) Electropherogram of PCR-amplified cDNA from an affected dog showing use of a cryptic splice site after disruption of the exon 20 donor site by the c.2363 +1 G >T variant. The result is a 40-bp truncation of the transcript. (C) Normal amino acid sequence; (D) The cryptic splice site is positioned at c.2324-c2325 and is predicted to lead to a sequence of five aberrant amino acids and premature termination (p.G775V$fs$*5).

be responsible for the disease (*Previtali, Quattrini & Bolino, 2007*; *Suter & Scherer, 2003*). In particular, variants of genes coding for the MTMR proteins and leading to abnormal myelin compaction have been described in CMT type 1, type 4B1 (variants in *MTMR2*) and type 4B2 (variant in *SBF2 (MTMR13)*, *MTMR2* regulatory binding partner) (*Previtali, Quattrini & Bolino, 2007*; *Houlden et al., 2001*; *Bolino et al., 2000*; *Conforti et al., 2004*; *Nelis et al., 2002*; *Senderek et al., 2003*; *Azzedine et al., 2003*; *Hirano et al., 2004*; *Luigetti et al., 2013*; *Murakami et al., 2013*; *Nouioua et al., 2011*; *Parman et al., 2004*; *Verny et al., 2004*; *Abuzenadah et al., 2013*; *He et al., 2018*; *Laššuthová et al., 2018*; *Negrao et al., 2014*; *Chen et al., 2014*). In this study we have mapped demyelinating neuropathy in the Miniature Schnauzer to canine chromosome 21 and have subsequently identified a genetic variant in and exon 19 of *SBF2* that shows strong segregation with the disease in this breed. Variants of the *SBF2* gene are known in humans to cause CMT type 4B2 (although a variant in exon 19 has not been found in humans). So far, thirty-seven humans with twenty-six pathogenic *SBF2* variants leading to CMT type 4B2 have been reported (*Baets et al., 2011*), a number that has dramatically increased recently with the discovery of nine new *SBF2* variants in one study (*Laššuthová et al., 2018*) and three new *SBF2* variants in another (*He et al., 2018*).

Myotubularins (MTM) form a family of 14 proteins (*Laporte et al., 2003*) that act as phosphoinositide phosphatase (PP) proteins and play a central role in neuromuscular

homeostasis (*Senderek et al., 2003*). Some are catalytically active (such as MTRM2) and dephosphorylate sub pools of phosphatidylinositols (PtdIns) (*Laporte et al., 2003*; *Biancalana et al., 2003*; *Blondeau et al., 2000*; *Laporte et al., 2001*; *Schaletzky et al., 2003*), while others are catalytically inactive and function as a scaffold to bind other myotubularins and proteins (such as MTMR13 binds MTMR2) (*Berger et al., 2006*; *Jean et al., 2012*). They are implicated in a number of physiological processes including cell proliferation, growth, survival, motility, cytoskeletal regulation and intracellular vesicle trafficking and signaling (endosomal-lysosomal pathways) (*Laporte et al., 2003*; *Buj-Bello et al., 2002*). Their importance is highlighted by the large number of diseases associated with variant of myotubularin genes (*Amoasii, Hnia & Laporte, 2012*; *McCrea & De Camilli, 2009*; *Nicot & Laporte, 2008*), but it remains poorly understood how these variants lead to disease.

Their ubiquitous expression, lack of functional redundancy and involvement in varied pathologies suggests that their expression and regulation finely varies between organs (*Kim et al., 2002*; *Laporte et al., 2002*). For example, in humans, a variant within the dDENN domain of *SBF2* leads to a demyelinating neuropathy (*Senderek et al., 2003*) and a variant within the DENN domain causes a thrombocytopenia without neurological manifestations (*Abuzenadah et al., 2013*). It is believed in this latter case that the genetic variant affects the solubility of the SBF2 protein but that the protein conformation is preserved (*Abuzenadah et al., 2013*).

In cases with involvement of the peripheral nerves, the analysis of the type of genetic variants associated with *SBF2* also provides a very good example of how variants in one gene, and within a particular domain, leads to different phenotypes, in particular with additional involvement of the eye. The first *SBF2* variant in humans was reported in a Turkish family and involved a deletion of 1,301 bp across exons 11 and 12, removing the unique dDENN motif (*Senderek et al., 2003*) but allowing the synthesis of a partially functional *SBF2* protein highlighting the possible role of dDENN in myelin compaction. This only caused a demyelinating neuropathy. Later, in a Tunisian and a Moroccan family, two nonsense variants in *SBF2* were identified in exons 23 (2875C>T (Gln956Stop)) and 27 (3586C>T (Arg1196Stop)) (*Azzedine et al., 2003*). These patients presented juvenile onset glaucoma associated with the demyelinating neuropathy, a phenotype that was also found in a Japanese family affected by another nonsense variant (1459C>T (Arg487Stop)) in exon 14 of *SBF2* gene (*Hirano et al., 2004*). These variants resulted in a truncated protein with deletion of the GRAM, myotube-related and PH domains and complete loss of function. Finally, with variants of the inactive phosphatase domain of *SBF2*, leading to presence of an abnormal but likely partially functional *SBF2* protein, human patients have a demyelinating peripheral neuropathy with or without glaucoma (*Luigetti et al., 2013*; *Laššuthová et al., 2018*). The occurrence of ocular anomalies in association with peripheral neuropathy is also present in the recent report of an inherited peripheral neuropathy in Black Russian terrier dogs (*Mhlanga-Mutangadura et al., 2016b*).

The finding of a variant in *SBF2* in the dogs of this report strikingly exemplifies the wide role of myotubularin proteins since a variant of *MTM1* has also been reported in a group of Labrador Retrievers, but leads to a centro-nuclear myopathy (*Beggs et al., 2010*), as it does in humans (*Kim et al., 2002*). These dogs are clinically normal at birth but

develop generalized muscular weakness and muscle atrophy at ~7 weeks of age and die in their first six months of life. *MTM1* is specifically involved in endosomal and membrane trafficking pathways, and late maturation and maintenance of T-tubules. In affected dogs, the structure of the T tubule network is affected, leading to disorganization of the muscle triads and their maintenance.

This new Miniature Schnauzer canine model allows useful comparisons between the human and canine disease for several reasons. First, this is a spontaneous (i.e., naturally occurring) model of a demyelinating neuropathy with focally folded myelin. This is important because the phenotype is likely more relevant and closer to the disease observed in humans, knowing the limitations of the rodent models. *SBF2* deficient mice have been created by inserting a gene-trap vector following exon 14 of *SBF2*, leading to lack of expression of the PH-GRAM and phosphatase domains and coiled-coil motif of *SBF2* (*Tersar et al., 2007*), which are the domains involved in binding, catalyzing and mediating interactions between active and inactive myotubularins (*Begley et al., 2006*; *Robinson & Dixon, 2005*). Although *SBF2* is broadly expressed in the mouse (*Kirfel et al., 2006*), the phenotype of this model differs from the naturally occurring disease since no reduction in the sciatic nerve conduction velocity was observed at 12 months of age in mice (this is also the case in a mouse model of the MTMR2 neuropathy (*Bonneick et al., 2005*)), which markedly contrasts with the profound slowing of nerve conduction velocity observed in dogs (*Houlden et al., 2001*) and humans (*Senderek et al., 2003*; *Azzedine et al., 2003*; *Hirano et al., 2004*; *Luigetti et al., 2013*). Clinically, these mice are not distinguishable from wild type animals but behavior analysis with the rotarod test identified deficient mice (*Tersar et al., 2007*).

Another group of researchers created a similar mouse mutant by inserting a gene-trap plasmid following exon 17 (*Robinson et al., 2008*), resulting in a similar defect as described above. Again, these animals did not show obvious behavioral abnormalities, but in contrast to the model from Tersar et al. (*Tersar et al., 2007*), showed reduced motor nerve conduction velocities (in the range of 20m/s). These two examples illustrate the complexity of establishing engineered animal models in the laboratory because the clinical phenotype obtained is not predictable, or is variable or lacking. In contrast, affected Miniature Schnauzer dogs present to veterinary hospitals with clear clinical signs, marked electrophysiological abnormalities and characteristic pathological features. As shown above, the variant described can be directly compared to variants affecting some human families, where protein abnormalities are more likely to be similar. Altogether, the signs observed in dogs provide means to follow-up the disease progression and to trial novel therapies. Secondly, the dog model could be used for *in vitro* studies, for example using fibroblast cultures that can be easily obtained from skin biopsies. In one study (*Ng et al., 2013*) using the Robinson's mouse model (*Robinson et al., 2008*), the consequences of *SBF2* variant on the endo-lysosomal pathway has been investigated and it was found that loss of *SBF2* does not dysregulate the endo-lysosomal membrane system in Schwann cells. Therefore, there is a need to further identify the impact of *SBF2* loss on the trafficking of specific cargo proteins. The prevalence of the disease in Miniature Schnauzers is currently not known but there is likely a pool of animals that could be available for tissue sampling

and further *in vitro* research. Future studies could focus on the identification of specific membrane proteins that may be abnormally trafficked in the absence of *SBF2*, as well as on defining how phosphatidylinositols are dysregulated by the loss of these proteins (*Dang et al., 2004*). Finally, it should be mentioned that, although spontaneous canine models are appealing, they come with limitations. In particular, their remain unknown as to how many cases could be accessed for research. The use of companion dogs in research requires strict ethical safeguards, and imposes limitation to access to tissue (including post-mortem) and interventions that can be performed. The cost of developing colonies of affected dogs within veterinary laboratories and cost of running clinical trial is also greater than the cost of working with rodents.

## CONCLUSIONS

In conclusion, we have identified a splicing variant in *SBF2* as the most likely cause for the demyelinating peripheral neuropathy observed in Miniature Schnauzer dogs. This provides the first naturally occurring large animal model of a demyelinating form of CMT disease, 4B2.

The discovery and description of naturally occurring models of CMT diseases open great research avenues. They render possible *in vitro* characterization of molecular pathways and functional work, such as with the use of fibroblasts obtained from the skin or nerve transcriptome analysis. The large pool of affected dogs and expansion of veterinary hospitals could allow collection of samples for this research and could pave the way for testing new therapies in canine models, as exemplified by the recent use of gene editing to restore dystrophin expression in a canine model of muscular dystrophy (*Amoasii et al., 2018*). This strategy could be applied to many genetic diseases, such as the demyelinating peripheral neuropathy presented here, while the use of a larger animal model allows real-life quantification of the benefit of proposed therapies.

## ACKNOWLEDGEMENTS

The authors are grateful to referring veterinarians, veterinary neurology specialists (Dr. Sofie Bhatti, Prof. Luc VanHam, Dr. An Vanhaesbrouck, Dr. Jérôme Couturier, Dr. Laurent Cauzinille, Prof. Diane Shelton) and to all dog owners and breeders who donated samples and shared pedigree data of their dogs. We thank the Dog Biomedical Variant Database Consortium (Gus Aguirre, Catherine André, Danika Bannasch, Doreen Becker, Brian Davis, Cord Drögemüller, Kari Ekenstedt, Kiterie Faller, Oliver Forman, Steve Friedenberg, Eva Furrow, Urs Giger, Christophe Hitte, Marjo Hytönen, Vidhya Jagannathan, Tosso Leeb, Hannes Lohi, Cathryn Mellersh, Jim Mickelson, Leonardo Murgiano, Anita Oberbauer, Sheila Schmutz, Jeffrey Schoenebeck, Kim Summers, Frank van Steenbeek, Claire Wade) and Natasha Olby for sharing whole genome sequencing data from control dogs. We also acknowledge all canine researchers who deposited dog whole genome sequencing data into public databases.

### Funding
The authors received no funding for this work.

### Competing Interests
Drs Sally Ricketts, Rebekkah Hitti and Oliver Forman are employed by the Animal Health Trust, and Dr Nicolas Granger is employed by Bristol Veterinary Specialists, CVS Referrals.

### Author Contributions
- Nicolas Granger, Sally Ricketts and Oliver Forman conceived and designed the experiments, performed the experiments, analyzed the data, contributed reagents/materials/analysis tools, prepared figures and/or tables, authored or reviewed drafts of the paper, approved the final draft.
- Alejandro Luján Feliu-Pascual analyzed the data, prepared figures and/or tables, approved the final draft.
- Charlotte Spicer performed the experiments, analyzed the data, prepared figures and/or tables, approved the final draft.
- Rebekkah Hitti performed the experiments, analyzed the data, prepared figures and/or tables, approved the final draft.
- Joshua Hersheson conceived and designed the experiments, performed the experiments, analyzed the data, prepared figures and/or tables, approved the final draft.
- Henry Houlden conceived and designed the experiments, analyzed the data, contributed reagents/materials/analysis tools, prepared figures and/or tables, authored or reviewed drafts of the paper, approved the final draft.

### Animal Ethics
The following information was supplied relating to ethical approvals (i.e., approving body and any reference numbers):

Genomic DNA from AHT control dogs utilised for this study were derived from buccal swab samples or from residual blood drawn for diagnostic veterinary purposes as part of the dog's veterinary care. Samples were taken following informed and written owner consent. Sample collection for genetic research has been approved by the Animal Health Trust Ethics Committee (24-2018E).

### DNA Deposition
The following information was supplied regarding the deposition of DNA sequences:

The variant has been archived by the European Variation Archive, Project: PRJEB34548, Analyses: ERZ1082320

### Data Availability
The primer sequences are available in the Supplementary File.

## Supplemental Information

Supplemental information for this article can be found online at http://dx.doi.org/10.7717/peerj.7983#supplemental-information.

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
