# Peer review of "Charcot-Marie-Tooth type 4B2 demyelinating neuropathy in miniature Schnauzer dogs caused by a novel splicing SBF2 (MTMR13) genetic variant: a new spontaneous clinical model"

_PeerJ, doi:10.7717/peerj.7983_

## Round 0.1 · original submission · Minor Revisions

Both reviewers have few minor comments and all these comments must be addressed before the paper can be accepted!

Reviewer 1 ·

Basic reporting

This is an interesting and well written study reporting a novel mutation in the SBF2 gene that cause a demyelinating neuropathy in miniature Schnauzer dogs.

In the introduction (lines 96-111) the authors review the canine genetic variants causing inherited polyneuropathies similar to CMT. It will be interesting for the readers if the authors enlarge the review and include also canine variants causing sensory and autonomic neuropathies. For example, the sensory-autonomic neuropathy in Border collies or the insensitivity to pain of hunting dogs ( Forman OP, Hitti RJ, Pettitt L, Jenkins CA, O'Brien DP, Shelton GD, De Risio L, Quintana RG, Beltran E, Mellersh C. An Inversion Disrupting FAM134B Is Associated with Sensory Neuropathy in the Border Collie Dog Breed. G3 (Bethesda). 2016 Sep 8;6(9):2687-92. doi: 10.1534/g3.116.027896. and Plassais J, Lagoutte L, Correard S, Paradis M, Guaguère E, Hédan B, Pommier A, Botherel N, Cadiergues MC, Pilorge P, Silversides D, Bizot M, Samuels M, Arnan C, Johnson R, Hitte C, Salbert G, Méreau A, Quignon P, Derrien T, André C. A Point Mutation in a lincRNA Upstream of GDNF Is Associated to a Canine Insensitivity to Pain: A Spontaneous Model for Human Sensory Neuropathies. PLoS Genet. 2016 Dec 29;12(12):e1006482. doi: 10.1371/journal.pgen.1006482.)

Experimental design

Experimental design is clear and the research is well within the aims and scope of the journal.

Validity of the findings

This represents the first report of a genetic variant in the SBF2 gene causing a naturally occurring polyneuropathy in miniature Schnauzers that is similar to CMT disease in humans. It represents a potential naturally occurring model for the human disease. I commend the authors for screening for the variant in such a large number of dogs of different breeds.

Additional comments

In the discussion the authors highlight all the advantages of using naturally occurring canine disease as a model for human diseases. It will be good to also include a paragraph mentioning the limitations of using canine diseases as models.

Reviewer 2 ·

Basic reporting

In this paper, Granger and colleagues describe the genetic mutation causing a demyelinating neuropathy in miniature Schnauzer dogs, previously described in another paper by the same authors. The paper is well-written, the genetic studies have been thoroughly performed despite the limited number of dogs. The discussion comparing the various phenotypes observed with different variants of the same gene is interesting and the literature review extensive. The figures are adequate and all necessary information has been provided. The paper is self-contained with a clear answer.

Experimental design

The paper fulfils the Aims and Scope of PeerJ. The research question is well-defined. The findings are novel and interesting. Investigation was rigorous using relevant techniques. No ethical issue has been identified.
All methods are thoroughly described.

Validity of the findings

As previously stated, all data have been provided and the results appear genuine and sounds. The conclusions are well stated.

Additional comments

- Line 250: for the muscle biopsy, you may want to refer to reference 21.
- Line 342: for consistency with line 344, name the mutations 2875C>T and 3586C>T.
- Figure 1: please also add a scale bar to inserts C and D.
- Figure 4B: please add the amino acid numbers to make it easier to the reader
- Figure 4B: if codon GGT in control/GTA in affected dog corresponds to amino acid 773, can you please clarify why the premature codon is at amino acid 746?

---

## Round 0.2 · accepted · Accept

Thanks for addressing all the points raised by reviewers.